# Recent Trends in Antisense Therapies for Duchenne Muscular Dystrophy

**DOI:** 10.3390/pharmaceutics15030778

**Published:** 2023-02-26

**Authors:** Harry Wilton-Clark, Toshifumi Yokota

**Affiliations:** Department of Medical Genetics, University of Alberta, Edmonton, AB T6G 2H7, Canada

**Keywords:** eteplirsen, golodirsen, viltolarsen, casimersen, WVE-N531, SRP-5051, DS-5141B, NS-089/NCNP-02, SCAAV9.U7.ACCA, ATL1102

## Abstract

Duchenne muscular dystrophy (DMD) is a debilitating and fatal genetic disease affecting 1/5000 boys globally, characterized by progressive muscle breakdown and eventual death, with an average lifespan in the mid–late twenties. While no cure yet exists for DMD, gene and antisense therapies have been heavily explored in recent years to better treat this disease. Four antisense therapies have received conditional FDA approval, and many more exist in varying stages of clinical trials. These upcoming therapies often utilize novel drug chemistries to address limitations of existing therapies, and their development could herald the next generation of antisense therapy. This review article aims to summarize the current state of development for antisense-based therapies for the treatment of Duchenne muscular dystrophy, exploring candidates designed for both exon skipping and gene knockdown.

## 1. Introduction

Duchenne muscular dystrophy (DMD) is a fatal genetic disease stemming from mutations in the *DMD* gene located on the X-chromosome [1,2]. This gene is responsible for the production of dystrophin, a protein expressed in tissues throughout the body, including the heart, skeletal muscles, smooth muscles, retina, and brain [1,3]. While our understanding of this protein is continuously evolving, dystrophin has been previously demonstrated to have myriad effects, including roles in retinal function, fetal neural development, synaptic signaling, metabolism, cognitive function, and satellite cell polarity and division [4,5,6,7,8]. Most notably, dystrophin is a crucial regulator of muscular strength and stability in muscle cells, where it complexes with various glycoproteins to form a scaffold connecting actin filaments to the sarcolemma [9]. In the absence of dystrophin, muscle fibers are prone to progressive degradation and fibrosis.

The *DMD* gene itself is the largest gene in the human genome, and myriad mutations are possible across its 79 exons that can result in DMD. The most common causative mutations are whole-exon deletions or duplications that disrupt the open reading frame (ORF) of *DMD* and lead to a non-functional protein product, accounting for 60–70% of all DMD cases [10,11]. The remainder of cases arise from smaller nonsense mutations, intronic mutations, frameshift-inducing indels, and UTR mutations. In the case of intronic mutations, they typically involve the deletion of an endogenous splice site or the introduction of a cryptic splice site, leading in both cases to an inappropriately spliced final mRNA product [11]. For UTR mutations, protein expression and function are altered via the dysregulation of mRNA translation, stability, and/or localization [12]. These mutations, particularly large exon deletions, tend to cluster in two mutational “hotspots” within *DMD* rather than occurring evenly across the gene. Approximately 73% of major deletions occur in the primary hotspot located between exons 43 and 55, with a smaller secondary hotspot encompassing 23% of major deletions between exons 2 and 22 [13].

Clinically, DMD affects approximately 1/5000 boys born worldwide [14,15,16]. Patients typically present within the first few years of life with lower limb weakness and difficulty rising from the floor without the aid of their arms, a finding known as Gower’s sign [17]. By their early teenage years, most patients will experience a total loss of ambulatory ability and require the use of a wheelchair [18,19,20]. Between the ages of 18 and 20, patients typically require ventilatory assistance as the disease begins to affect the respiratory and cardiac muscles. This progression ultimately proves fatal for most patients, with DMD-related cardiomyopathy as the leading cause of death and a median lifespan of only 28 years [21]. No cure currently exists for DMD.

The current standard of care for DMD is usually corticosteroid treatment, which has been shown to delay the progression of disease and prolong ambulation [22]. However, this approach fails to address the underlying cause of DMD and is associated with numerous side effects, including weight gain, bone weakness, and the potential for adrenal insufficiency with prolonged use [23].

More recently, a novel treatment approach known as antisense therapy has gained traction, with four antisense-based drugs for DMD gaining FDA approval since 2016. Antisense therapy is a generalized term for therapeutics relying on small nucleic acid analogues called antisense oligonucleotides (AONs). These AONs are designed to hybridize with targeted mRNA, at which point expression of the mRNA can be altered through numerous mechanisms, including splice switching, gene knockdown, gene upregulation, or alteration of polyadenylation [24,25,26,27,28]. For the treatment of DMD, the most common approaches are AON-mediated splice switching and gene knockdown (Figure 1). The mechanism of different AONs varies based on the specific chemistry of the AON and the target site [24,25]. If the AON is designed with oligonucleotides susceptible to ribonuclease H, such as endogenous DNA, ribonuclease is recruited to degrade the mRNA bound by the AON, leading to a reduction in the abundance of mRNA transcripts and a corresponding reduction in the protein product [29,30]. This approach is selected when knockdown of the targeted gene is desirable to alleviate a pathogenic phenotype. In contrast, splice switching involves using oligonucleotides with ribonuclease-resistant chemistry that do not induce ribonuclease H, such as phosphorodiamidate morpholinos, that target regions on the pre-mRNA transcript that regulate splicing [25,31]. Rather than inducing degradation, these AONs interfere with the binding and interaction of splice nucleic acids and proteins, manipulating splicing to promote the production of the desired final mRNA product [32,33,34]. In the case of DMD, AONs are typically targeted to splice or splice enhancer sites in order to stimulate the therapeutic inclusion or exclusion of a target exon such that the normal reading frame is restored to the protein.

Almost all current antisense therapies for DMD use the splice switching approach, aiming to selectively skip *DMD* exons adjacent to a mutation in order to restore the reading frame of dystrophin [35,36]. The product of this skipped transcript is an internally deleted but partially functional dystrophin protein. The same truncated protein is naturally produced in the much milder Becker muscular dystrophy and inducing the production of this protein has been demonstrated to impart benefits in both preclinical and clinical studies [35,37,38,39]. This approach, often referred to as exon skipping therapy (EST), has seen massive popularity for DMD in recent years. Four AON candidates have received conditional FDA approval for the treatment of DMD, and many more remain in clinical and preclinical trials. The purpose of this article is to provide an overview of these drugs, summarizing the current status of antisense therapy for the treatment of DMD.

## 2. Current AONs in Development for DMD

### 2.1. FDA Approval Obtained

At the time of writing, four AONs targeting three different exons have received conditional FDA approval for the treatment of DMD: eteplirsen (Sarepta, exon 51 skipping), viltolarsen (NS pharma, exon 53), golodirsen (Sarepta, exon 53), and casimersen (Sarepta, exon 45) [37,40,41,42]. All four drugs are phosphorodiamidate morpholino oligomers (PMOs), a class of synthetic AONs containing a six-sided morpholine ring and phosphorodiamidate backbone which is resistant to enzymatic degradation. While all four candidates have successfully completed initial phase III clinical trials, continued approval will be contingent upon long-term clinical benefit, which will be assessed via ongoing longitudinal phase III trials.

These longitudinal studies are expected to conclude in April 2024 for golodirsen and casimersen, November 2024 for eteplirsen, and December 2024 for viltolarsen [43,44,45]. As the most advanced antisense candidates for DMD, these studies will be pivotal for guiding the future of antisense therapy for DMD patients. The results from these studies will determine not only whether each currently approved AON will retain FDA approval, but also whether antisense and exon skipping therapy in general are suitable for the long-term treatment of DMD. Particularly for golodirsen, these results will aim to dispute previous results from a phase II extension study of golodirsen, which found no significant improvement in clinical outcomes following three years of golodirsen treatment compared to natural history controls [46].

### 2.2. Phase II Clinical Trials

While no AON candidates other than the aforementioned PMOs with conditional FDA approval have entered Phase III trials, six different candidates are in varying stages of Phase II trials: ATL1102 (Antisense Therapeutics), SCAAV9.U7.ACCA (Astellas Pharma), SRP-5051 (Sarepta), NS-089/NCNP-02 (NS Pharma), WVE-N531 (Wave Life Sciences), and DS-5141B (Daiichi Sankyo).

#### 2.2.1. ATL1102

ATL1102 is the most advanced of these candidates, having successfully completed Phase IIa trials in Australia [47]. Unlike other AONs for DMD which aim to restore dystrophin production, ATL1102 is instead focused on treating the inflammation associated with DMD. ATL1102 was designed as a gapmer with a central ribonuclease-inducing DNA core and flanking 2′-O-(2-methoxyethyl) oligos targeted to CD49d mRNA, which encodes for a key subunit of the human very-late antigen (VLA4) associated with inflammation [48]. By recruiting ribonucleases to degrade CD49d transcripts, ATL1102 thus aims to reduce inflammation and inflammation-mediated tissue damage in DMD [49].

Results from the completed Phase II trials found that a weekly injection of ATL1102 was generally safe in patients, with no serious treatment-related adverse effects after 24 weeks of treatment [50]. No significant changes were found in blood lymphocyte count, a marker of inflammation, throughout the study, although the authors did note a significant increase in CD3 + CD49d + T lymphocytes 4 weeks after stopping the treatment. Further, no changes in grip strength or pinch strength were noted throughout the study [50]. Based on the favorable safety data and apparent stabilizing effect of ATL1102, the authors expressed optimism in the continued development of this drug. However, no details regarding further trials are yet available.

#### 2.2.2. SCAAV9.U7.ACCA

SCAAV9.U7.ACCA is another unique antisense candidate which aims to treat patients containing a single-exon duplication of *DMD* exon 2. This therapy strays away from direct AON treatment, instead opting for treatment with an adeno-associated virus (AAV) vector which allows for the endogenous production of non-coding U7 small nuclear RNAs (U7snRNAs) coupled to a sequence targeting exon 2 [51]. These U7snRNAs act in a similar fashion to AONs, inducing exon 2 skipping by targeting splice donor and acceptor sites on the pre-mRNA flanking exon 2 [52]. This leads to the production of normal, full-length dystrophin transcripts and proteins.

Preclinical studies in mice identified that SCAAV9.U7.ACCA was able to effectively skip one of the duplicate copies of exon 2, restoring the reading frame and enabling dystrophin production with extremely high efficacy nearing wild-type levels of dystrophin [52,53]. The researchers also found that both copies of exon 2 are occasionally skipped, resulting in an out-of-frame dystrophin transcript [54]. Interestingly, these transcripts still express functional dystrophin through an internal ribosomal entry site located on exon 5, so no correction was needed to account for this occurrence.

Phase I/II clinical trials are currently underway for SCAAV9.U7.ACCA, with an expected completion date of November 2025 [55]. This study will provide data regarding the safety of SCAAV9.U7.ACCA, as well as preliminary measures of efficacy and ability to restore dystrophin production. Notably, the safety of AAV-based therapies has been called into question recently following the tragic deaths of multiple patients in independent clinical trials assessing AAV-mediated drugs for DMD and X-linked myotubular myopathy [56,57,58]. Therefore, the safety findings from this study will be particularly important in determining whether the development of this therapy can proceed.

#### 2.2.3. SRP-5051

A major limitation of existing PMO-based therapies is that PMOs exhibit poor cellular uptake and are rapidly cleared through the renal system [59]. This reduces the bioavailability and efficacy of PMO treatments, thus requiring a higher dose to confer a therapeutic benefit. To address these limitations, recent research has explored the use of PMOs conjugated to cell-penetrating peptides, known as PPMOs [60,61]. By promoting cellular uptake, PPMOs are designed to be more effective than their unmodified PMO counterparts.

The most clinically advanced PPMO candidate is SRP-5051, a PPMO targeting exon 51 that is currently in phase II clinical trials [62]. SRP-5051, also known as vesleteplirsen, targets the same patient population as eteplirsen but with the added benefit of a proprietary cell-penetrating peptide moiety, and can therefore be thought of as a next-generation eteplirsen. As the first PPMO in clinical trials, validating the safety of SRP-5051 is of the utmost importance, and phase II trials were placed on clinical hold from June to September 2022 following an instance of treatment-related hypomagnesemia [62,63]. Despite this setback, phase II trials are expected to conclude in August 2024, and will provide information on the safety, pharmacokinetics, and preliminary efficacy of dystrophin restoration for SRP-5051.

#### 2.2.4. WVE-N531

WVE-N531 is an AON targeting exon 53 designed using Wave Life Sciences’ proprietary phosphoryl guanidine-containing (PN) backbone [64]. Preclinical studies exploring this novel chemistry identified that compared to oligos with standard phosphorothioate or phosphodiester chemistries, PN oligos demonstrate improved efficacy, pharmacokinetics, and activity in difficult-to-reach brain tissues [65,66].

WVE-N531 is nearing the end of its phase I/II trials, with results expected in December 2022. These trials will primarily provide information regarding the safety and pharmacokinetics of WVE-N531, with secondary findings exploring preliminary efficacy as measured by dystrophin restoration [64]. While no formal results are available yet, an interim press release from Wave indicates that WVE-N531 demonstrated a positive safety profile across all tested doses after six weeks of treatment, and that despite low dystrophin levels, effective exon skipping was observed at the mRNA level [67].

#### 2.2.5. NS-089/NCNP-02

Developed by NS Pharma, NS-089/NCNP-02 is the first AON candidate aiming to induce skipping of exon 44 [68,69]. The chemistry used for this drug is unspecified, though it is reasonable to assume that it may use a similar PMO chemistry to NS Pharma’s previous AON candidate, viltolarsen. Unlike AONs in development for exons 45, 51, and 53 that must compete with the previously approved AONs for each exon, NS-089/NCNP-02 would become the only available treatment for mutations amenable to exon 44 skipping if approved.

NS-089/NCNP-02 is currently in a phase II extension study following the successful completion of phase I/II trials. A press release regarding the phase I/II study from NS Pharma identified that no serious adverse effects occurred during the trial which required discontinuation, and that dystrophin was effectively restored in a dose-dependent manner after treatment with NS-089/NCNP-02 [70]. Results from the extended phase II trial will provide further data regarding safety, dystrophin restoration, and clinical efficacy, and are expected in July 2023 [69].

#### 2.2.6. DS-5141B

The final candidate in phase II development is DS-5141B, an AON designed by Daiichi-Sankyo to skip exon 45 constructed using 2′-O,4′-C-ethylene-bridged nucleic acid (ENA) oligos [71,72]. Compared to PMOs, ENAs were reported to demonstrate increased exon skipping efficiency in both skeletal and cardiac tissue in mouse studies, which is of particular importance considering the high rate of cardiomyopathy-related mortality in DMD [71]. The authors hypothesize that this improvement is due to improved binding affinity with dystrophin pre-mRNA, however the mechanism is not well-understood yet.

Following positive safety findings in phase I/II trials, DS-5141B is currently undergoing a phase II extension study which will further validate the safety of this AON, as well as provide preliminary clinical efficacy data assessing both skeletal muscle and cardiac phenotype [72,73,74]. The results from this study are expected in March 2023.

### 2.3. Preclinical/Phase I Clinical Trials

The majority of the newer DMD antisense therapies being developed are focused on AON conjugates, aiming to address the limitations associated with standard, unmodified AONs. Most of these conjugations involve proprietary technology designed to boost efficacy, which will seek to disrupt the unmodified PMOs that have previously received FDA approval. However, considering the early stage of development for many of these therapies, specific details are often unavailable.

#### 2.3.1. PGN-EDO51

PGN-EDO51 (PepGen) is currently in the phase I stage of clinical testing. This oligo aims to skip *DMD* exon 51 and is constructed with PepGen’s Enhanced Delivery Oligonucleotide (EDO) technology, which is a proprietary cell-penetrating PPMO, like Sarepta’s SRP-5051 [75]. While the available information surrounding PGN-EDO51 is minimal, a press release by PepGen claims that Phase I testing in healthy normal volunteers showed a favorable safety profile and high levels of exon 51 skipping [75]. Accordingly, PepGen will aim to initiate phase IIa trials in 2023. It is worth noting that press releases can be heavily biased, and that conclusions regarding safety and efficacy cannot be properly drawn until more information is available.

#### 2.3.2. ENTR-601-44

Another PPMO, ENTR-601-44, is also aiming to initiate clinical trials in the near future [76]. Created by Entrada Therapeutics using their Endosomal Escape Vehicle (EEV) platform, this PPMO is specifically designed to promote endosomal escape by inducing the collapse of endosomes which contain entrapped PPMO [77,78,79]. This approach improves the bioavailability of the conjugated AON, thus improving efficacy [80]. Entrada is in the process of submitting an Investigational New Drug (IND) application to the FDA, however a press release revealed that this may be delayed by a clinical hold imposed by the FDA in December 2022 [76]. Further details regarding this therapy may become available once ENTR-601-44 enters formalized clinical testing.

Taking a slightly different approach, both Dyne Therapeutics and Avidity Biosciences have recently unveiled antibody-conjugated AONS hybridized to transferrin receptor 1 (TfR1) antibodies [81,82]. Preclinical studies assessing this technology in mice have identified that TfR1-AONS demonstrate significantly increased dystrophin expression and functional improvement compared to unmodified PMO [81].

#### 2.3.3. DYNE-251

Dyne’s candidate, DYNE-251, aims to use this technology to skip *DMD* exon 51. Phase I trials have recently begun with a planned extension to phase II, which will assess safety, dystrophin restoration, and clinical improvement after 24 and 120 weeks of treatment [83]. This study is expected to conclude in November 2026.

#### 2.3.4. AOC 1044

AOC 1044 is Avidity’s candidate and aims to treat patients with DMD amenable to exon 44 skipping. Avidity announced via press release that they were commencing clinical trials for AOC 1044 in October 2022, although at the time of writing no information is publicly available in the FDA database [84]. The trial is named EXPLORE44 and will assess the safety and preliminary efficacy of AOC 1044 treatment. A summary of all antisense candidates mentioned is available in Table 1.

## 3. Future Directions

Finally, although there are no candidates in the clinical developmental pipeline yet, multi-exon skipping is an approach that has the potential to transform the field of antisense therapy for the treatment of DMD [85]. Rather than the single-exon approach of current AONs, multi-exon skipping aims to skip multiple exons simultaneously with an AON “cocktail” [86]. This method improves the applicability of antisense therapy for DMD, and a cocktail skipping exons 45–55 has been validated in vitro and in vivo which could be applicable to nearly 50% of all DMD patients [86,87,88]. Furthermore, this research has demonstrated that spontaneous skipping can occur during multi-exon skipping, which reduces the number of AONs required in the cocktail, reducing treatment complexity and cost [89]. At this time, the development of multi-exon skipping cocktails is limited by regulatory hurdles, as each AON in the cocktail must undergo separate clinical approval [41].

Additional AON chemistries are also being explored at the preclinical stage, which could eventually be incorporated into clinical studies if they demonstrate improved safety or efficacy characteristics compared to existing AONs. One noteworthy approach is exploring the use of tricyclo-DNA (tcDNA), a conformationally constrained nucleic acid analog that demonstrates effective uptake in many tissues and penetration of the blood–brain barrier, a limitation of some other AONs [90,91,92]. Further, tcDNA has been found to be effective at sizes as small as 13 nt, which could help to alleviate AON-induced toxicity at higher doses [93]. Mouse studies have found that tcDNA AONs can effectively restore dystrophin production in the dystrophic mouse model, but also demonstrated AON-induced kidney toxicity [93]. While tcDNA could be an exciting future avenue, further studies are required before it reaches the clinical space.

Although the focus of this article emphasized antisense therapy, numerous other therapeutic options are also under development. Microdystrophin therapy, which seeks to treat DMD by supplementing patients with an AAV vector encoding for a truncated but functional “microdystrophin” protein, has multiple candidates in phase III clinical trials and could soon join antisense therapies as FDA-approved DMD treatments [94,95,96,97]. Another approach which has shown some promise is readthrough therapy, which aims to treat DMD arising from premature termination codons [98,99]. Readthrough therapy has gained approval in Europe and seeks to also gain FDA approval with ongoing longitudinal phase III trials [100]. More recently, cell-based therapies have also been under development and have demonstrated notable benefit in cardiac tissue, which is a key weakness of current antisense therapies [101,102]. While further research is required to determine which therapies will prove most beneficial for patients, each approach—antisense or otherwise—helps to bring a functional treatment for DMD closer to the patients who need it most.

## Figures and Tables

**Figure 1 pharmaceutics-15-00778-f001:**
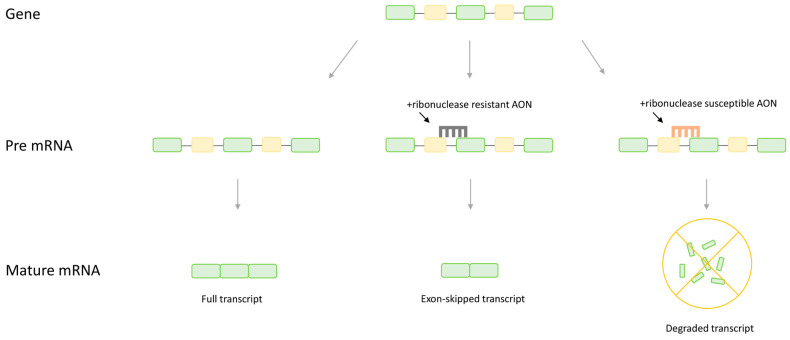
A comparison of the effect of different antisense oligonucleotide (AON) chemistries on the final mRNA transcript. Ribonuclease-resistant AONs targeted to splice sites will induce splice switching, in this case leading to removal of an exon form the mature mRNA. Ribonuclease-susceptible AONs will recruit ribonuclease H and induce hydrolysis, eliminating the transcript. Note, AONs are typically 15–25 nt in length and are not to scale.

**Table 1 pharmaceutics-15-00778-t001:** An overview of all antisense therapies currently in clinical development, organized by their therapeutic target.

Therapeutic Target	Name	AON Chemistry	Sponsor	Status
Exon 53 Skipping	viltolarsen	PMO	NS Pharma	Conditionally Approved
	golodirsen	PMO	Sarepta Therapeutics	Conditionally Approved
	WVE-N531	phosphoryl guanidine (PN) backbone	Wave Life Sciences	Phase I/II
Exon 51 Skipping	eteplirsen	PMO	Sarepta Therapeutics	Conditionally Approved
	SRP-5051	PPMO	Sarepta Therapeutics	Phase II
	PGN-EDO51	PPMO	PepGen	Phase I
	DYNE-251	Antibody-PMO	Dyne Therapeutics	Phase I
Exon 45 Skipping	casimersen	PMO	Sarepta Therapeutics	Conditionally Approved
	DS-5141B	2′-O,4′-C-ethylene-bridged nucleic acid (ENA)	Daiichi Sankyo	Phase II
Exon 44 Skipping	NS-089/NCNP-02	Unknown	NS Pharma	Phase II
	AOC 1044	Antibody-PMO	Avidity Biosciences	Phase I
	ENTR-601-44	PPMO	Entrada Therapeutics	Preclinical
Exon 2 Skipping	SCAAV9.U7.ACCA	AAV U7snRNA	Astellas Pharma	Phase I/II
CD49d Knockdown	ATL1102	2′-O-(2-methoxyethyl)	Antisense Therapeutics	Phase IIa

## Data Availability

Not applicable.

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
