# Peer review of "Recent Trends in Antisense Therapies for Duchenne Muscular Dystrophy"

_pharmaceutics, 2023, doi:10.3390/pharmaceutics15030778_

Round 1

Reviewer 1 Report

The review of Wilton-Clark and Yokoto gives a compact and comprehensible overview of current antisense therapies for the severe Duchenne muscular dystrophy disease. The manuscript is well written and addresses the actual status of AON-based drugs for DMD as well as limitations of this therapeutic option in a satisfying way.

Author Response

Reviewer 1:

The review of Wilton-Clark and Yokoto gives a compact and comprehensible overview of current antisense therapies for the severe Duchenne muscular dystrophy disease. The manuscript is well written and addresses the actual status of AON-based drugs for DMD as well as limitations of this therapeutic option in a satisfying way.

Author response: Thank you for taking the time to read our manuscript, and for the favourable review.

Reviewer 2 Report

The authors are providing a brief, in parts very superficial update of antisense therapies for Duchenne MD.

There are several parts that need either further details, clarification or in some parts correction, and I will mention these in order thru the manuscript rather than priority.

L8 Cited incidence... 1/3500 is an old figure.  I was under the impression that this had been revised to approx 1/5000 ... please check some other references

L25.  The dystrophin roles are still evolving.  There is now evidence that dystrophin plays crucial roles in satellite cell division and proliferation/division

L35.  Please expand on the UTR mutations.  I would guess that these would be less common than missense mutations (very rare but nevertheless still present) and what about small frame-shifting in/dels

L53.  I am not sure if AONs are small DNA-like molecules.... nucleic acid analogues would be more appropriate

L56.  there are far more mechanisms that splice switching or gene knock-down.  Gene up-regulation, altered polyadenylation.  

L58. the DNA AON is not degraded by RNaseH but the RNA strand  ... this needs to be reworked

P63 Splice switching involved AONs that are nuclease resistant but do not induce RNaseH activity,  The description here is for exon skipping AONs but splice switching is a more general term and would involve targeting intronic/exonic silence elements.

L65.  It is more complex that simple steric blocking... otherwise simple targeting of canonical acceptor or donor splice sites should disrupt  splicing

Figure 1.  Rework.  Otherwise it looks like the AONs are targeting the entire intron with end/begining of flanking exons

L75. I would have thought functionality of induced dystrophin (after exon skipping) would have been demonstrated in the milder allelic form of Becker MD 

L113.  Incorrect.  ATL1102 is a gapmer.  MOE ends and a DNA core to induce RNaseH degradation of CD49d mRNA.  An AON with all MOE bases would not support RNaseH but could induce exon skipping

L132 Mention that the U7 snRNA is coupled to a sequence targeting exon 2.

L134. Perhaps expand that this approach is only relevant to exon 2 duplications.  A duplication of 2-5 or 2-10 would not respond to single exon 2 skipping and make a normal protein.

L150. perhaps more details on how more effective.  ie dosing frequency is monthly (rather than weekly) and more protein is reported

L196. Which company is involved with DS-5141B

L218. Not sure if it is even worth reporting press releases , especially ones that show a :favourable safety profile with high levels of exon skipping from healthy volunteers.  This was a single amending dose and generally well tolerated.  I question what safety issues can be addressed from a single dose.  The authors mention minimal data is available but could expand on such company hype.

Table 1.. Perhaps mention that the "approved" drugs only have conditional or accelerated approval and further trials /testing are ongoing

Author Response

Reviewer 2:

The authors are providing a brief, in parts very superficial update of antisense therapies for Duchenne MD. There are several parts that need either further details, clarification or in some parts correction, and I will mention these in order thru the manuscript rather than priority.

Author response: Thank you for taking the time to read our manuscript, and for the insightful feedback. Please find each point addressed below.

L8 Cited incidence... 1/3500 is an old figure.  I was under the impression that this had been revised to approx 1/5000 ... please check some other references

Author response: The prevalence has been updated to reflect the most recent studies (~1/5000), which have been cited.

L25.  The dystrophin roles are still evolving.  There is now evidence that dystrophin plays crucial roles in satellite cell division and proliferation/division

Author response: This section has been expanded to briefly mention the other roles of dystrophin in the body. It now reads “This gene is responsible for the production of dystrophin, a protein expressed in tissues throughout the body including the heart, skeletal muscles, smooth muscles, retina, and brain [1,3]. While our understanding of this protein is continuously evolving, dystrophin has been previously demonstrated to have myriad effects including roles in retinal function, fetal neural development, synaptic signaling, metabolism, cognitive function, and satellite cell polarity and division [4–8].  Most notably, dystrophin is a crucial regulator of muscular strength and stability in muscle cells, where it complexes with various glycoproteins to form a scaffold connecting actin filaments to the sarcolemma [9].”

L35.  Please expand on the UTR mutations.  I would guess that these would be less common than missense mutations (very rare but nevertheless still present) and what about small frame-shifting in/dels

Author response: The description of smaller mutation types has been expanded on. It now reads: The remainder of cases arise from smaller nonsense mutations, intronic mutations, frameshift-inducing indels, and UTR mutations. In the case of intronic mutations, they typically involve the deletion of an endogenous splice site or the introduction of a cryptic splice site, leading in both cases to an inappropriately spliced final mRNA product [11]. For UTR mutations, protein expression and function is altered via the dysregulation of mRNA translation, stability, and/or localization [12].

L53.  I am not sure if AONs are small DNA-like molecules.... nucleic acid analogues would be more appropriate

Author response: Thank you for the suggestion. “Nucleic acid analogues” has been used as more precise language.

L56.  there are far more mechanisms that splice switching or gene knock-down.  Gene up-regulation, altered polyadenylation.  

Author response: Additional mechanisms used less frequently for DMD have been added to make this section more comprehensive.

L58. the DNA AON is not degraded by RNaseH but the RNA strand  ... this needs to be reworked

Author response: This has been reworded to prevent misinterpretation. It now reads “If the AON is designed with oligonucleotides susceptible to ribonuclease H, such as endogenous DNA, ribonuclease is recruited to degrade the mRNA bound by the AON, leading to a reduction in the abundance of mRNA transcripts and a corresponding reduction in protein product”

P63 Splice switching involved AONs that are nuclease resistant but do not induce RNaseH activity,  The description here is for exon skipping AONs but splice switching is a more general term and would involve targeting intronic/exonic silence elements.

Author response: This section has been updated to better reflect the mechanisms less commonly used for DMD. It now reads “These AONs are designed to hybridize with targeted mRNA at which point expression of the mRNA can be altered through numerous mechanisms including splice switching, gene knockdown, gene up-regulation, or alteration of polyadenylation [23–27]. For the treatment of DMD, the most common approaches are AON-mediated splice switching and gene knockdown”

L65.  It is more complex that simple steric blocking... otherwise simple targeting of canonical acceptor or donor splice sites should disrupt  splicing

Author response: This section has been updated to more generally reflect interactions with splice machinery.

Figure 1.  Rework.  Otherwise it looks like the AONs are targeting the entire intron with end/begining of flanking exons

Author response: The figure caption now mentions that AONs are typically 15-25nt in length and not to scale.

L75. I would have thought functionality of induced dystrophin (after exon skipping) would have been demonstrated in the milder allelic form of Becker MD 

Author response: This is a true point, however exon skipping does not lead to ~100% truncated protein production (as in BMD) and so additional justification of the clinical benefit of EST is required. This section has been updated and now reads “The product of this skipped transcript is an internally deleted but partially functional dystrophin protein. The same truncated protein is naturally produced in the much milder Becker muscular dystrophy, and inducing the production of this protein has been demonstrated to impart benefit in both pre-clinical and clinical studies”

L113.  Incorrect.  ATL1102 is a gapmer.  MOE ends and a DNA core to induce RNaseH degradation of CD49d mRNA.  An AON with all MOE bases would not support RNaseH but could induce exon skipping

Author response: This section has been reworded to prevent miscommunication. It now reads: “ATL1102 was designed as a gapmer with a central ribonuclease-inducing DNA core and flanking 2′-O-(2-methoxyethyl) oligos targeted to CD49d mRNA, which encodes for a key subunit of the human very late antigen (VLA4) associated with inflammation [40]”

L132 Mention that the U7 snRNA is coupled to a sequence targeting exon 2.

Author response: This has been updated to mention the targeting of exon 2 when u7 snRNA is introduced.

L134. Perhaps expand that this approach is only relevant to exon 2 duplications.  A duplication of 2-5 or 2-10 would not respond to single exon 2 skipping and make a normal protein.

Author response: This section now introduces with “SCAAV9.U7.ACCA is another unique antisense candidate which aims to treat patients containing a single-exon duplication of DMD exon 2”

L150. perhaps more details on how more effective.  ie dosing frequency is monthly (rather than weekly) and more protein is reported

Author response: SRP-5051 is also a monthly dosing schedule. The high efficacy of dystrophin restoration in pre-clinical studies was added. It now reads “Pre-clinical studies in mice identified that SCAAV9.U7.ACCA was able to effectively skip one of the duplicate copies of exon 2, restoring the reading frame and enabling dys-trophin production with extremely high efficacy nearing wild-type levels of dystrophin”.

L196. Which company is involved with DS-5141B

Author response: Daiichi Sankyo has been mentioned in the manuscript to match the information available in table 1.

L218. Not sure if it is even worth reporting press releases , especially ones that show a :favourable safety profile with high levels of exon skipping from healthy volunteers.  This was a single amending dose and generally well tolerated.  I question what safety issues can be addressed from a single dose.  The authors mention minimal data is available but could expand on such company hype.

Author response: This section has been updated to better discuss potential bias in company press releases.

Table 1.. Perhaps mention that the "approved" drugs only have conditional or accelerated approval and further trials /testing are ongoing

Author response: Thank you for your point. Conditionally approved drugs have been labelled as such to improve accuracy.

Reviewer 3 Report

The paper is well written and it is always useful to have updated papers describing the state of the art about new therapies. This is even more true for DMD at the moment, with new gene therapies approaching. The major concern about this paper is its lacking of information. Indeed some strategies are only cited without describing in deep their characteristics. Others are not cited at all as the AAV9-U7 exon 51 (U7ex51) or the tricyclo-DNA (tcDNA). Some punctuations need to be reviewed.

Author Response

Reviewer 3:

The paper is well written and it is always useful to have updated papers describing the state of the art about new therapies. This is even more true for DMD at the moment, with new gene therapies approaching. The major concern about this paper is its lacking of information. Indeed some strategies are only cited without describing in deep their characteristics. Others are not cited at all as the AAV9-U7 exon 51 (U7ex51) or the tricyclo-DNA (tcDNA). Some punctuations need to be reviewed.

Author response: Thank you for taking the time to read our manuscript, and for the insightful feedback. More information has been added throughout based on your and other reviewer’s feedback. The focus of this review was on antisense options currently under clinical development, and many technologies which are only at the pre-clinical stage were deemed out of scope. TcDNA has now been discussed in the future directions section.

Reviewer 4 Report

Very nice and concise review of the current state of affairs for antisense therapies development for DMD.  I enjoyed reading this very much.

Author Response

Reviewer 4:

Very nice and concise review of the current state of affairs for antisense therapies development for DMD.  I enjoyed reading this very much.

Author response: Thank you for taking the time to read our manuscript, and for the favourable review.

Reviewer 5 Report

pharmaceutics-2162120

Recent Trends in Antisense Therapies for Duchenne muscular dystrophy

Harry Wilton-Clark and Toshifumi Yokota

This manuscript well summarizes the recent antisense therapies for DMD. This is an informative review for the clinicians and basic scientists.

I have some suggestions to improve the manuscript.

1) It would be nicer if the authors could provide antisense oligo sequences and could schematically show where they bind in Dystrophin pre-mRNAs.

2) For SCAAV9.U7.ACCA, some people are not familiar with U7 snRNP. It would be better to add structure of U7snRNA with antisense region for Dystrophin gene.

Basically, it would be wonderful if the authors provide more Figures as information.

Author Response

Reviewer 5:

 This manuscript well summarizes the recent antisense therapies for DMD. This is an informative review for the clinicians and basic scientists. I have some suggestions to improve the manuscript. Basically, it would be wonderful if the authors provide more Figures as information.

Author response: Thank you for taking the time to read our manuscript, and for the insightful feedback. Please find your points of concern addressed below.

  • It would be nicer if the authors could provide antisense oligo sequences and could schematically show where they bind in Dystrophin pre-mRNAs.

Author response: Unfortunately, the oligo sequence for many of the AONs discussed is unavailable, and therefore showing their exact binding site is not feasible. We have aimed to organize table 1 by the exon which each AON targets to provide this information.

  • For SCAAV9.U7.ACCA, some people are not familiar with U7 snRNP. It would be better to add structure of U7snRNA with antisense region for Dystrophin gene.

Author response: Our goal with this review was to provide an easily digestible overview of AONs currently undergoing clinical testing for DMD. Many different antisense chemistries are mentioned in this review including U7 snRNA, PMO, PPMO, phosphoryl guanidine backbone, antibody-PMO, ENA, and MOE. Numerous excellent review articles exist which show the varying structures and properties of these chemistries, however the authors believe it is outside the scope of this review.